# Link prediction in Hypergraphs using Graph convolutional networks

## Abstract

Link prediction in simple graphs is a fundamental problem in which new links between nodes are predicted based on the observed structure of the graph. However, in many real-world applications, there is a need to model relationships among nodes which go beyond pairwise associations. For example, in a chemical reaction, relationship among the reactants and products is inherently higher-order. Additionally, there is need to represent the direction from reactants to products. Hypergraphs provide a natural way to represent such complex higher-order relationships. Even though Graph Convolutional Networks (GCN) have recently emerged as a powerful deep learning-based approach for link prediction over *simple* graphs, their suitability for link prediction in *hypergraphs* is unexplored – we fill this gap in this paper and propose Neural Hyperlink Predictor (NHP). NHP adapts GCNs for link prediction in hypergraphs. We propose two variants of NHP – NHP-U and NHP-D – for link prediction over undirected and directed hypergraphs, respectively. To the best of our knowledge, NHP-D is the first method for link prediction over directed hypergraphs. Through extensive experiments on multiple real-world datasets, we show NHP's effectiveness.

## 1 Introduction

The problem of link prediction in graphs has numerous applications in the fields of social network analysis (Liben-Nowell & Kleinberg, 2003), knowledge bases (Nickel et al., 2016), bioinformatics (Lü & Zhou, 2011) to name a few. However, in many real-world problems relationships go beyond pairwise associations. For example, in chemical reactions data the relationship representing a group of chemical compounds that can react is inherently higher-order and similarly, the co-authorship relationship in a citation network is higher-order etc. Hypergraphs provide a natural way to model such higher-order complex relations. Hyperlink prediction is the problem of predicting such missing higher-order relationships in a hypergraph.

Besides the higher-order relationships, modeling the direction information between these relationships is also useful in many practical applications. For example, in the chemical reactions data, in addition to predicting groups of chemical compounds which form reactants and/or products, it is also important to predict the direction between reactants and products, i.e., a group of reactants react to give a group of products. Directed hypergraphs (Gallo et al., 1993) provide a way to model the direction information in hypergraphs. Similar to the undirected hypergraphs, predicting the missing hyperlinks in a directed hypergraph is also useful in practical settings. Figure 1 illustrates the difference between modeling the chemical reactions data using undirected and directed hypergraphs. Most of the previous work on hyperlink prediction (Zhou et al., 2006; Zhang et al., 2018) focus only on undirected hypergraphs. In this work we focus both on undirected and directed hypergraphs.

Recently, Graph Convolutional Networks (GCNs) (Kipf & Welling, 2017) have emerged as a powerful tool for representation learning on graphs. GCNs have also been successfully applied for link prediction on normal graphs (Zhang & Chen, 2018; van den Berg et al., 2018; Schlichtkrull et al., 2018; Kipf & Welling, 2016). Inspired by the success of GCNs for link prediction in graphs and deep learning in general Wang et al. (2017), we propose a GCN-based framework for hyperlink prediction which works for both undirected and directed hypergraphs. We make the following contributions:

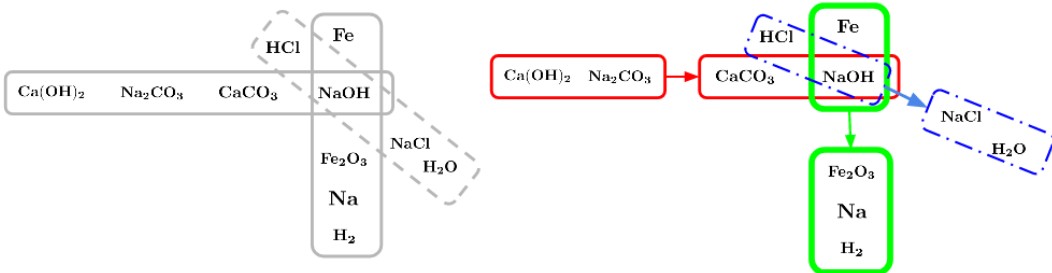

Figure 1: Illustrating the difference between modeling chemical reactions data using undirected and directed hypergraphs. To the left is the undirected hypergraph, in which both the reactants and products are present in the same hyperlink. Whereas in the directed hypergraph (to the right), for a given reaction, the reactants are connected by one hyperlink and products are connected by another hyperlink and both these hyperlinks are connected by a directed link.

- We propose a Graph Convolutional Networks (GCN)-based framework called Neural Hyperlink Predictor (NHP) for the problem of hyperlink prediction. To the best of our knowledge, this is the first ever deep learning based approach for this problem.
- We extend the proposed NHP for the problem of hyperlink prediction in directed hypergraphs. To the best of our knowledge, this is the first ever attempt at the problem of link prediction in directed hypergraphs.
- Through extensive experiments on multiple real-world datasets, we show the effectiveness of proposed NHP for link prediction in both undirected and directed hypergraphs.

We have released NHP's source code at this anonymous location: `https://anonymous.4open.science/repository/7d86231e-f6ba-4795-ae51-ac28d89f1521/`.

## 2 RELATED WORK

In this section, we briefly review related work in deep learning on graphs and link prediction on hypergraphs.

**Learning representations on graphs**: The key advancements in learning low-dimensional node representations in graphs include matrix factorisation-based methods, random-walk based algorithms, and deep learning on graphs (Hamilton et al., 2017). Our work is based on deep learning on graphs.

*Geometric deep learning* (Bronstein et al., 2017) is an umbrella phrase for emerging techniques attempting to generalise (structured) deep neural network models to non-Euclidean domains such as graphs and manifolds. The earliest attempts to generalise neural networks to graphs embed each node in an Euclidean space with a recurrent neural network (RNN) and use those embeddings as features for classification or regression of nodes or graphs (Gori et al., 2005; Scarselli et al., 2009).

A CNN-like deep neural neural network on graphs was later formulated in the *spectral domain* in a pioneering work (Bruna et al., 2014) by a mathematically sound definition of convolution on graph employing the analogy between the classical Fourier transforms and projections onto the eigen basis of the graph Laplacian operator (Hammond et al., 2011). Initial works proposed to learn smooth spectral multipliers of the graph Laplacian, although at high computational cost (Bruna et al., 2014; Henaff et al., 2015). To resolve the computational bottleneck and avoid the expensive computation of eigenvectors, the ChebNet framework (Defferrard et al., 2016) learns Chebyshev polynomials of the graph Laplacian (hence the name ChebNet). The graph convolutional network (GCN) (Kipf & Welling, 2017) is a simplified ChebNet framework that uses simple filters operating on 1-hop local neighborhoods of the graph.

A second formulation of convolution on graph is in the *spatial domain* (or equivalently in the vertex domain) where the localisation property is provided by construction. One of the first formulations of a spatial CNN-like neural network on graph generalised standard molecular feature extraction methods based on circular fingerprints (Duvenaud et al., 2015). Subsequently, all of the above types

(RNN, spectral CNN, spatial CNN on graph) were unified into a single message passing neural network (MPNN) framework (Gilmer et al., 2017) and a variant of MPNN has been shown to achieve state-of-the-art results on an important molecular property prediction benchmark.

The reader is referred to a comprehensive literature review (Bronstein et al., 2017) and a survey (Hamilton et al., 2017) on the topic of deep learning on graphs and learning representation on graphs respectively. Below, we give an overview of related research in link prediction on hypergraphs where relationships go beyond pairwise.

**Link Prediction on hypergraphs**: Machine learning on hypergraphs was introduced in a seminal work (Zhou et al., 2006) that generalised the powerful methodology of spectral clustering to hypergraphs and further inspired algorithms for hypergraph embedding and semi-supervised classification of hypernodes.

Link prediction on hypergraph (hyperlink prediction) has been especially popular for social networks to predict higher-order links such as *a user releases a tweet containing a hashtag* (Li et al., 2013) and to predict metadata information such as tags, groups, labels, users for entities (images from Flickr) (Arya & Worring, 2018). Techniques for hyperlink prediction on social networks include ranking for link proximity information (Li et al., 2013) and matrix completion on the (incomplete) incidence matrix of the hypergraph (Arya & Worring, 2018; Monti et al., 2017). Hyperlink prediction has also been helpful to predict multi-actor collaborations (Sharma et al., 2014).

In other works, a dual hypergraph has been constructed from the intial (primal) hypergraph to cast the hyperlink prediction as an instance of vertex classification problem on the dual hypergraph (Lugo-Martinez & Radivojac, 2017). Coordinated matrix maximisation (CMM) predicts hyperlinks in the adjacency space with non-negative matrix factorisation and least square matching performed alternately in the vertex adjacency space (Zhang et al., 2018). CMM uses expectation maximisation algorithm for optimisation for hyperlink prediction tasks such as predicting missing reactions of organisms' metabolic networks.

## 3 PRELIMINARIES

**Undirected hypergraph** is an ordered pair $H = (V, E)$ where $V = \{v_1, \cdots, v_n\}$ is a set of $n$ hypernodes and $E = \{e_1, \cdots, e_m\} \subseteq 2^V$ is a set of $m$ hyperlinks.

The problem of link prediction in an incomplete undirected hypergraph $H$ involves predicting missing hyperlinks from $\bar{E} = 2^V - E$ based on the current set of observed hyperlinks $E$. The number of hypernodes in any given hyperlink $e \in E$ can be any integer between $1$ and $2^n$. This variable cardinality of a hyperlink makes traditional graph-based link prediction methods infeasible because they are based on exactly two input features (those of the two nodes potentially forming a link). The variable cardinality problem also results in an exponentially large inference space because the total number of potential hyperlinks is $O(2^n)$. However, in practical cases, there is no need to consider all the hyperlinks in $\bar{E}$ as most of them can be easily filtered out (Zhang et al., 2018). For example, for the task of finding missing metabolic reactions, we can restrict hyperlink prediction to all feasible reactions because the infeasible reactions seldom have biological meanings. In other cases such as predicting multi-author collaborations of academic/technical papers, hyperlinks have cardinalities less than a small number, as papers seldom have more than 6 authors. The number of restricted hyperlinks in such practical cases is not exponential and hence hyperlink prediction on the restricted set of hyperlinks becomes a feasible problem.

Formally, a hyperlink prediction problem (Zhang et al., 2018) is a tuple $(H, \mathcal{E})$, where $H = (V, E)$ is a given incomplete hypergraph and $\mathcal{E}$ is a set of (restricted) candidate hyperlinks with $E \subseteq \mathcal{E}$. The problem is to find the most likely hyperlinks missing in $H$ from the set of hyperlinks $\mathcal{E} - E$.

**Directed hypergraph** (Gallo et al., 1993) is an ordered pair $H = (V, E)$ where $V = \{v_1, \cdots, v_n\}$ is a set of $n$ hypernodes and $E = \{(t_1, h_1), \cdots, (t_m, h_m)\} \subseteq 2^V$ is a set of $m$ directed hyperlinks. Each $e \in E$ is denoted by $(t, h)$ where $t \subseteq V$ is the *tail* and $h \subseteq V$ is the *head* with $t \neq \Phi$, $h \neq \Phi$. As shown in figure 1, chemical reactions can be modeled by directed hyperlinks with chemical substances forming the set $V$. Observe that this model captures and is general enough to subsume previous graph models:

- an undirected hyperlink is the special case when $t = h$

- a directed simple link (edge) is the special case when $|t| = |h| = 1$

Similar to the undirected case, the directed hyperlink prediction problem is a tuple $(H, \mathcal{E})$, where $H = (V, E)$ is a given incomplete directed hypergraph and $\mathcal{E}$ is a set of candidate hyperlinks with $E \subseteq \mathcal{E}$. The problem is to find the most likely hyperlinks missing in $H$ from the set of hyperlinks $\mathcal{E} - E$.

# 4 PROPOSED FRAMEWORK NHP

In this section we discuss the proposed approach for link prediction in hypergraphs. Our proposed NHP can predict both undirected and directed hyperlinks in a hypergraph. We start with how NHP can be used to predict undirected hyperlinks and then explain how NHP can be extended to the directed case.

## 4.1 NHP-U: LINK PREDICTION IN UNDIRECTED HYPERGRAPHS

Given an undirected hypergraph $H = (V, E)$, as a first step NHP constructs a dual hypergraph $H^* = (V^*, E^*)$ of $H$ defined as follows.

**Hypergraph Duality (Scheinerman & Ullman, 2011)** The *dual* hypergraph of a hypergraph $H = (V, E)$ with $V = \{v_1, \ldots, v_n\}$ and $E = \{e_1, \ldots, e_m\}$, denoted by $H^* = (V^*, E^*)$ is obtained by taking $V^* = E$ as the set of hypernodes and $E^* = \{e_1^*, \ldots, e_n^*\}$ such that $e_i^* = \{e \in E : v_i \in e\}$ with $e_i^*$ corresponding to $v_i$ for $i = 1, \ldots, n$. The vertex-hyperedge incidence matrices of $H$ and $H^*$ are transposes of each other.

The problem of link prediction in $H$ can be posed as a binary node classification problem in $H^*$ (Lugo-Martinez & Radivojac, 2017). A label $+1$ on a node in $H^*$ indicates the presence of a hyperlink in $H$ and a label of $-1$ indicates the absence. For the problem of semi-supervised node classification on the dual hypergraph $H^*$, we use Graph Convolutional Networks (GCN) on the graph obtained from the clique expansion of $H^*$. Clique expansion (Zhou et al., 2006; Feng et al., 2019) is a standard and a simple way of approximating a hypergraph into a planar graph by replacing every hyperlink of size $s$ with an $s$-clique (Feng et al., 2018).

**Graph Convolutional Network (Kipf & Welling, 2017)** Let $\mathcal{G} = (\mathcal{V}, \mathcal{E})$, with $N = |\mathcal{V}|$, be a simple undirected graph with the adjacency matrix $A \in \mathbb{R}^{N \times N}$, and let the data matrix be $X \in \mathbb{R}^{N \times p}$. The data matrix has $p$-dimensional real-valued vector representations for each node in the graph. The forward model for a simple two-layer GCN takes the following simple form:

$$Z = f_{GCN}(X, A) = \text{softmax}\left( \bar{A} \ \text{ReLU}\left( \bar{A} X \Theta^{(0)} \right) \Theta^{(1)} \right), \tag{1}$$

where $\bar{A} = \tilde{D}^{-\frac{1}{2}} \tilde{A} \tilde{D}^{-\frac{1}{2}}$, $\tilde{A} = A + I$, and $\tilde{D}_{ii} = \sum_{j=1}^{N} \tilde{A}_{ij}$. $\Theta^{(0)} \in \mathbb{R}^{p \times h}$ is an input-to-hidden weight matrix for a hidden layer with $h$ hidden units and $\Theta^{(1)} \in \mathbb{R}^{h \times r}$ is a hidden-to-output weight matrix. The softmax activation function is defined as $\text{softmax}(x_i) = \frac{\exp(x_i)}{\sum_j \exp(x_j)}$ and applied row-wise.

For semi-supervised multi-class classification with $q$ classes, we minimise the cross-entropy error over the set of labeled examples, $\mathcal{V}_L$.

$$\mathcal{L} = -\sum_{i \in \mathcal{V}_L} \sum_{j=1}^{q} Y_{ij} \ln Z_{ij}. \tag{2}$$

The weights of the graph convolutional network, viz. $\Theta^{(0)}$ and $\Theta^{(1)}$, are trained using gradient descent.

We note that Lugo-Martinez & Radivojac (2017) also follow a similar approach of constructing dual graph followed by node classification for hyperlink prediction. However, ours is a deep learning based approach and (Lugo-Martinez & Radivojac, 2017) do not perform link prediction in the directed hypergraphs.

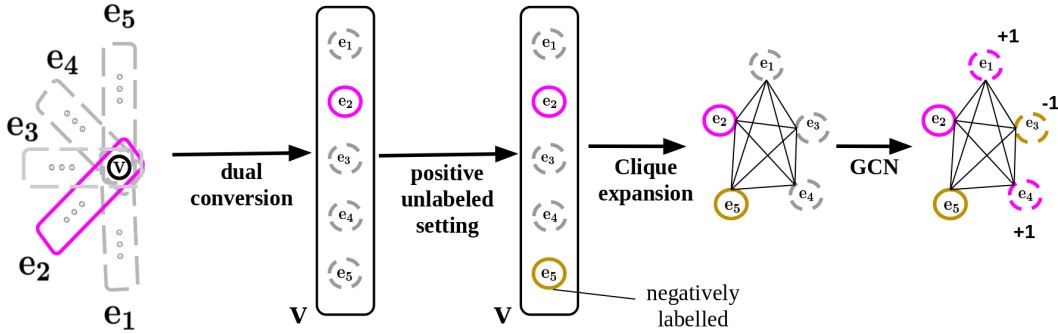

Figure 2: (best seen in colour) The proposed NHP framework. We convert the hypergraph with the observed hyperlinks and the candidate hyperlinks into its dual in which hyperlinks are converted into hypernodes. We then use a technique from positive unlaballed learning to get plausible negatively labelled hypernodes. The clique expansion of the hypergraph is used to approximate the hypergraph. Then a GCN is run on the graph to classify the unlabelled hypernodes. A label of $+1$ on $e_1$ indicates presence of the $e_1$ in the primal. For more details, refer to section 4.

**Learning on hypergraphs in the positive unlabelled setting** The cross-entropy objective of GCN in equation 2 inherently assumes the presence of labeled data from at least two different classes and hence cannot be directly used for the positive unlabeled setting. A challenging variant of semi-supervised learning is positive-unlabelled learning (Elkan & Noto, 2008) which arises in many real-world applications such as text classification, recommender systems, etc. In this setting, only a limited amount of positive examples are available and the rest are unlabelled examples.

In positive unlabelled learning framework, we construct a plausible negative sampling set based on data similarity (Han & Shen, 2016). We calculate the average similarity of each unlabelled point $u \in \mathcal{E} - E$, to the positive labelled examples in $E$:

$$s_u = \frac{1}{|E|} \sum_{e \in E} S\bigg(f(u), f(e)\bigg) \tag{3}$$

where $S$ represents a similarity function and $f$ represents a map that maps an example to a $d$-dimensional emebedding space $f : \mathcal{E} \to \mathbb{R}^d$. We then rank the unlabelled training examples in $\mathcal{E} - E$ in the ascending order of their average similarity scores. We then select the top ones in the ascending order (i.e. with the lowest similarity values) to construct the set of plausible negative examples $F \subseteq \mathcal{E} - E$. The intuition here is that the set of plausible negative examples contain examples most dissimilar to the positive examples. The GCN on the hypergraph can subsequently be run by minimising the objective 2 over the positive examples in $E$ and the plausible negative examples in $F$ i.e. $\mathcal{V}_L = E \cup F$.

## 4.2 NHP-D: LINK PREDICTION IN DIRECTED HYPERGRAPHS

As explained in Section 3, hyperlink prediction in directed hypergraphs is the problem of predicting directing links which are tail-head set pairs. However, in practice the collection of the tail and head sets is also incomplete. Therefore, the problem of link prediction in directed hypergraphs also requires to predict missing tail and head sets of nodes besides predicting directed links among them. The tail and head sets of nodes can be thought of as undirected hyperlinks and the directed hyperlink is between these undirected hyperlinks. A straight forward approach for this problem would be to predict the undirected hyperlinks first and then followed by predicting the directed links between pairs. However, this sequential approach may not produce desirable results as the error during the training for directed part would not have any impact on the training for the undirected part. Therefore, we propose the following joint learning scheme,

$$\mathcal{L}_{joint} = \mathcal{L}_u + \mathcal{L}_d = \bigg( - \sum_{i \in \mathcal{V}_L} \sum_{j=1}^{2} Y_{ij} \ln Z_{ij} \bigg) + \bigg( - \sum_{(i,j) \in \mathcal{W}_L} \sum_{k=0}^{1} d_{ijk} \ln D_{ijk} \bigg). \tag{4}$$

| dataset | iAF692 | iHN637 | iAF1260b | iJO1366 |
|---|---|---|---|---|
| # substances, $\|V\|$ | 628 | 698 | 1668 | 1805 |
| # actual reactions, $\|E\|$ | 690 | 785 | 2388 | 2583 |
| # candidate reactions, $\|\mathcal{E}\|$ | 2406 | 3050 | 7260 | 7672 |

Table 1: Statistics of the four metabolic networks used as undirected hypergraphs

| dataset | CORA | DBLP |
|---|---|---|
| # authors, $\|V\|$ | 1072 | 685 |
| # actual papers (collaborations), $\|E\|$ | 2708 | 1590 |
| # candidate papers, $\|\mathcal{E}\|$ | 5416 | 3180 |
| # features (vocabulary size), $p$ | 1433 | 602 |

Table 2: Statistics of the two coauthorship networks used as undirected hypergraphs

We denote $\mathcal{W}_L^+ = \{(t, h) \in E\}$ to be a set of positively labelled directed pairs in loss $\mathcal{L}_d$. In other words $d_{ij1} = 1$ and $d_{ij0} = 0$ for all $(i, j) \in \mathcal{W}_L^+$. The tail hyperlink and the corresponfding head hyperlink are separate hypernodes in the dual and form directed hyperlinks in the primal (with the direction from $t$ to $h$). The set $\mathcal{W}_L^+$ consists of directed hyperlinks that currently exist in the given directed hypergraph. Note that, for loss $\mathcal{L}_u$, the set of positively labelled hypernodes will be,

$$\mathcal{V}_L^+ = \bigcup_{(t,h) \in W_L^+} t \cup h. \tag{5}$$

We sample $|\mathcal{W}_L^+| = |E|$ hypernodes (in the dual) from the unlabelled data using the positive unlabelled approach of 3 to get the set of $\mathcal{W}_L^-$ pairs. We label these pairs negative i.e. $d_{ij1} = 0$ and $d_{ij0} = 1$ for all $(i, j) \in \mathcal{W}_L^-$. We construct $\mathcal{V}_L^- = \bigcup_{(t,h) \in W_L^-} t \cup h$ similarly as in 5. The sets $\mathcal{V}_L = \mathcal{V}_L^+ \cup \mathcal{V}_L^-$ and $\mathcal{W}_L = \mathcal{W}_L^+ \cup \mathcal{W}_L^-$ are used to minimise the objective 6.

To explain how $D$ is computed, we rewrite equation 1 as:

$$Z = f_{GCN}(X, A) = \text{softmax}\left(\bar{A} \ \text{ReLU}\left(\bar{A} X \Theta^{(0)}\right) \Theta^{(1)}\right) = \text{softmax}(\mathcal{X}), \tag{6}$$

We use $D = g(x_1, x_2)$ with $g$ being a function that takes the dual hypernode representations $x_1 \in \mathcal{X}$ and $x_2 \in \mathcal{X}$ and is parameterised by for example a simple neural network. In the experiments, we used a simple 2-layer multilayer perceptron on the concatenated embeddings $x_1 || x_2$ i.e.

$$g(x_1, x_2) = \text{MLP}_\Theta\left(x_1 || x_2\right).$$

We train $\Theta^{(0)}$, $\Theta^{(1)}$, and $\Theta$ end-to-end using backpropagation.

## 5 EXPERIMENTS FOR UNDIRECTED HYPERLINK PREDICTION

In this section, we evaluate NHP on hyperlink prediction in undirected hypergraphs. We performed two different sets of experiments whose motivations and setups are as follows.

**Predicting reactions of metabolic networks:** Reconstructed metabolic networks are important tools for understanding the metabolic basis of human diseases, increasing the yield of biologically engineered systems, and discovering novel drug targets. We used four datasets of (Zhang et al., 2018) and we show the statistics of the datasets used in table 1. For each dataset, we randomly generated fake reactions according to the substance distribution of the existing reactions. So, the candidate reactions contain already existing ones and the randomly generated fake ones. The number of fake reactions generated is equal to the number of already existing ones. Given a small number of reactions the task is to predict the other reactions.

| dataset | iAF692 | iHN637 | iAF1260b | iJO1366 | CORA | DBLP |
|---|---|---|---|---|---|---|
| SHC (Zhou et al., 2006) | 0.69 | 0.69 | 0.69 | 0.70 | 0.61 | 0.64 |
| node2vec (Grover & Leskovec, 2016) | 0.72 | 0.73 | **0.71** | 0.70 | 0.59 | 0.61 |
| CMM (Zhang et al., 2018) | 0.50 | 0.54 | 0.58 | 0.60 | 0.63 | 0.44 |
| GCN on star expansion (Ying et al., 2018) | 0.56 | 0.54 | 0.53 | 0.48 | 0.54 | 0.42 |
| NHP-U (ours) | **0.75** | **0.75** | **0.71** | **0.72** | **0.64** | **0.65** |

Table 3: mean AUC (higher is better) over 10 trials. NHP achieves consistently superior performance over its baselines for all the datasets. Refer to section 5 for more details.

| dataset | iAF692 | iHN637 | iAF1260b | iJO1366 | CORA | DBLP |
|---|---|---|---|---|---|---|
| SHC (Zhou et al., 2006) | $248 \pm 6$ | $289 \pm 4$ | $1025 \pm 4$ | $1104 \pm 19$ | $1056 \pm 14$ | $845 \pm 18$ |
| node2vec (Grover & Leskovec, 2016) | $299 \pm 10$ | $303 \pm 4$ | $1100 \pm 13$ | $1221 \pm 21$ | $1369 \pm 15$ | $813 \pm 9$ |
| CMM (Zhang et al., 2018) | $170 \pm 6$ | $225 \pm 10$ | $827 \pm 1$ | $963 \pm 15$ | $1452 \pm 13$ | $651 \pm 20$ |
| GCN on star expansion (Ying et al., 2018) | $174 \pm 5$ | $219 \pm 12$ | $649 \pm 10$ | $568 \pm 18$ | $1003 \pm 14$ | $646 \pm 15$ |
| NHP-U (ours) | $\mathbf{313 \pm 6}$ | $\mathbf{360 \pm 5}$ | $\mathbf{1258 \pm 9}$ | $\mathbf{1381 \pm 9}$ | $\mathbf{1476 \pm 20}$ | $\mathbf{866 \pm 15}$ |
| # missing hyperlinks, $\|\Delta E\|$ | 621 | 706 | 2149 | 2324 | 2437 | 1431 |
| Recall@ $\|\Delta E\|$ for NHP-U | 0.50 | 0.51 | 0.58 | 0.59 | 0.60 | 0.61 |

Table 4: mean ($\pm$ std) number of hyperlinks recovered over 10 trials (higher is better) among the top ranked $\|\Delta E\|$ hyperlinks. NHP achieves consistently superior performance over its baselines for all the datasets. Refer to section 5 for more details.

**Predicting multi-author collaborations in coauthorship networks:** Research collaborations in scientific community have been extensively studied to understand team dynamics in social networks (Newman, 2001; Barabási et al., 2002; Bai et al., 2018; Chen et al., 2015). Coauthorship data provide a means to analyse research collaborations. We used cora and dblp for coauthorship data. The statistics are shown in table 2 and the construction of the datasets is pushed to the supplementary. A coauthorship hypergraph (primal) contains each author as a hypernode and each paper represents a hyperlink connecting all the authors of the paper. The corresponding dual hypergraph considers each author as a hyperlink connecting all the papers (hypernodes) coauthored by the author. The hyperlink prediction problem is given a small number of collaborations, to essentially predict other collaborations among a given set of candidate collaborations.

## 5.1 EXPERIMENTAL SETUP

For each dataset we randomly sampled $10\%$ of the hypernodes to get $E$ and then we sampled an equal number of negatively-labelled hypernodes from $\mathcal{E} - E$ in the positive-unlabelled setting of 3. To get the feature matrix $X$, we used random $32-$dimensional Gaussian features ($p = 32$ in equation 1) for the metabolic networks and bag-of-word features shown in table 2 for the coauthorship datasets. For fake papers we generated random Gaussian bag-of-word features. We used node2vec (Grover & Leskovec, 2016) to learn low dimensional embedding mapping, $f : \mathcal{E} \to \mathbb{R}^d$ with $d = 128$. We used the clique expansion of the dual hypergraph as input graph to node2vec and cosine similarity to compute similarity between two embeddings.

We compared NHP against the following state-of-the-art baselines for the same $E$ as constructed above.

- **Spectral hypergraph Clustering (SHC) (Zhou et al., 2006)**: SHC outputs classification scores by $f = (I - \xi\Theta)^{-1}y$. We used SHC on the dual hypergraph.
- **node2vec (Grover & Leskovec, 2016)**: One of the most popular node embedding approaches. We note that (Grover & Leskovec, 2016) have shown node2vec to be superior to DeepWalk (Perozzi et al., 2014) and LINE (Tang et al., 2015) and hence we compared against only node2vec. We used node2vec to embed the nodes of the clique expansion (of the dual). We then used an MLP on the embeddings with the semi-supervised objective of equation 2 in the positive unlabelled setting of equation 3.
- **Co-ordinated Matrix Maximisation (CMM) (Zhang et al., 2018)**: The matrix factorisation-based CMM technique uses the EM algorithm to determine the presence or absence of candidate hyperlinks.
- **GCN on star expansion (Ying et al., 2018)**: PinSage (Ying et al., 2018) is a GCN-based method designed to work on the (web-scale) bipartite graph of Pinterest. The Pinterest

graph can be seen as the star expansion of a hypergraph (Agarwal et al., 2006) with pins (hypernodes) on one side of the partition and boards (hyperlinks) on the other side. We have compared NHP against star expansion. This is essentially approximating the hypergraph with its star exapansion and then running a GCN over it (instead of the clique expansion of NHP).

## 5.2 EXPERIMENTAL RESULTS

Similar to (Zhang et al., 2018), we report mean AUC over 10 trials in table 3 and the mean number of hyperlinks recovered over 10 trials in the top ranked $|\Delta E|$ ones in table 4. Note that $\Delta E \subset E$ is the set of missing hyperlinks with $|\Delta E| = 0.9 * |E|$. As we can observe, we consistently outperform the baselines in both the metrics. We believe this is because of the powerful non-linear feature extraction capability of GCNs. We also report Recall@ $|\Delta E|$ for NHP-U for all datasets (to make the numbers across datasets somewhat comparable). It is got through dividing the mean number of hyperlinks recovered by $\Delta E$.

## 6 EXPERIMENTS FOR DIRECTED HYPERLINK PREDICTION

| dataset | iAF692 | iHN637 | iAF1260b | iJO1366 |
|---|---|---|---|---|
| # substances, $|V|$ | 595 | 668 | 1542 | 1665 |
| # actual reactions | 519 | 571 | 1544 | 1683 |
| # candidate reactions, $|\mathcal{E}|$ | 917 | 1160 | 2531 | 2669 |

Table 5: Statistics of the four metabolic networks used as directed hypergraphs

| dataset | iAF692 | iHN637 | iAF1260b | iJO1366 |
|---|---|---|---|---|
| node2vec (Grover & Leskovec, 2016) + MLP | 0.53 | **0.52** | 0.58 | 0.56 |
| CMM (Zhang et al., 2018) + MLP | 0.53 | **0.52** | 0.52 | 0.51 |
| GCN on star expansion (Ying et al., 2018) + MLP | 0.48 | **0.52** | 0.53 | 0.50 |
| NHP-D (sequential) | 0.57 | 0.48 | **0.63** | **0.60** |
| NHP-D (joint) | **0.58** | 0.51 | 0.62 | 0.59 |

Table 6: mean AUC over 10 trials for all the datasets. Both the proposed models achieve similar results. Refer to section 6 for more details.

| dataset | iAF692 | iHN637 | iAF1260b | iJO1366 |
|---|---|---|---|---|
| node2vec (Grover & Leskovec, 2016) + MLP | $255 \pm 5$ | $237 \pm 5$ | $838 \pm 13$ | $902 \pm 11$ |
| CMM (Zhang et al., 2018) + MLP | $253 \pm 9$ | $\mathbf{241 \pm 11}$ | $757 \pm 26$ | $848 \pm 21$ |
| GCN on star expansion (Ying et al., 2018) + MLP | $242 \pm 5$ | $\mathbf{241 \pm 10}$ | $786 \pm 13$ | $852 \pm 11$ |
| NHP-D (sequential) | $\mathbf{263 \pm 7}$ | $221 \pm 10$ | $867 \pm 31$ | $\mathbf{954 \pm 29}$ |
| NHP-D (joint) | $262 \pm 8$ | $236 \pm 8$ | $\mathbf{869 \pm 13}$ | $944 \pm 20$ |
| # missing hyperlinks, $|\Delta E|$ | 467 | 514 | 1390 | 1683 |
| Recall@ $|\Delta E|$ for NHP-D (joint) | 0.56 | 0.46 | 0.63 | 0.56 |

Table 7: mean ($\pm$ std) number of hyperlinks recovered over 10 trials (higher is better) among the top ranked $|\Delta E|$ hyperlinks. Both the proposed models achieve similar results. Refer to section 6 for more details.

We used the same four metabolic networks to construct directed hyperlinks. The metabolic reactions are encoded by stoichiometric matrices. The negative entries in a stoichiometric matrix indicate reactants and positive entries indicate products. We extracted only those reactions which have at least two substances in each of the reactant side and the product side and the statistics are shown in table 5. We labelled randomly sampled 10% of the hyperlinks in the data and use the remaining 90% unlabelled data for testing. Tables 6 and 7 show the results on the datasets. NHP-D (joint) is the model proposed in 4.2. On the other hand, NHP-D (sequential) is the model which treats undirected hyperlink prediction and direction prediction separately. NHP-D (sequential) first runs a GCN on the clique expansion of the undirected hypergraph to get the node embeddings (without softmax) and then runs a multi-layer perceptron on the concatenated emebeddings to predict the directions.

| dataset | iAF692 | iHN637 | iAF1260b | iJO1366 | CORA | DBLP |
|---|---|---|---|---|---|---|
| random negative sampling | 0.72 | 0.76 | 0.77 | 0.77 | 0.62 | 0.58 |
| positive-unlabeled learning | 0.75 | 0.75 | 0.71 | 0.72 | 0.64 | 0.65 |
| mixed | 0.78 | 0.79 | 0.78 | 0.79 | 0.74 | 0.69 |

Table 8: mean AUC (higher is better) over 10 trials. Mixed consistently achieves the best AUC values. It provides benefits of both positive unlabeled learning and random negative sampling. Refer to section 7 for more details.

| dataset | iAF692 | iHN637 | iAF1260b | iJO1366 | CORA | DBLP |
|---|---|---|---|---|---|---|
| random negative sampling | $236 \pm 32$ | $415 \pm 47$ | $967 \pm 125$ | $1074 \pm 168$ | $978 \pm 181$ | $543 \pm 125$ |
| positive-unlabeled learning | $313 \pm 6$ | $360 \pm 5$ | $1258 \pm 9$ | $1381 \pm 9$ | $1476 \pm 20$ | $866 \pm 15$ |
| mixed | $317 \pm 11$ | $481 \pm 13$ | $1202 \pm 29$ | $1336 \pm 34$ | $1496 \pm 62$ | $813 \pm 36$ |
| # missing hyperlinks, $|\Delta E|$ | 621 | 706 | 2149 | 2324 | 2437 | 1431 |

Table 9: mean ($\pm$ std) number of hyperlinks recovered over 10 trials among the top ranked $|\Delta E|$ hyperlinks. Positive unlabeled learning of 3 achieves consistently lower standard deviations than the the other two. The standard deviations of random negative sampling are on the higher side. Refer to section 7 for more details.

We compared against the following baselines with a multi-layer perceptron (MLP):

- **node2vec (Grover & Leskovec, 2016) + MLP**: We used node2vec for the undirected hyperlink prediction part (as explained in section 5 and a 2-layer perceptron to predict the direction between hyperlinks with the joint objective of equation 4.

- **Co-ordinated Matrix Maximisation (CMM) (Zhang et al., 2018) + MLP**: The matrix factorisation-based CMM technique uses the EM algorithm to determine the presence or absence of candidate hyperlinks for the following optimisation problem:

$$\min_{\Lambda, W} ||A + U \Lambda U^T - W W^T||_F^2$$

$$\text{subject to } \lambda_i \in \{0, 1\}, i = 1, \cdots, m$$

$$W \geq 0$$

We used a $2-$layer perceptron in a sequential manner. To get a representation for a hyperlink $e$, we took the mean of the representations of all the hypernodes $v \in e$ from the matrix $W$ above. Note CMM works on the primal hypergraph.

- **GCN on star expansion (Ying et al., 2018) + MLP**: We used the star expansion to approximate the input hypergraph and then run a GCN on it (instead of the clique expansion of NHP). A 2-layer perceptron is used to predict the direction between hyperlinks with the joint objective of equation 4.

As we see in the tables, both NHP-D (joint) and NHP-D (sequential) perform similarly. This can be attributed to the fact that training data to predict directions between hyperlinks is sparse and hence the learned hypernode representations of both the models are similar. Please note that existing apporaches for link prediction on directed simple graphs cannot be trivially adopted for this problem because of the sparsity in the training data.

**Comparison to baselines** NHP-D outperforms the baselines on 3 out of 4 datasets. The dataset iHN637 seems to be a very challenging dataset on which each model recovers less than half the number of missing hyperlinks.

## 7 COMPARISON OF POSITIVE UNLABELED LEARNING VS. NEGATIVE SAMPLING UNIFORMLY AT RANDOM

In order to justify positive-unlabeled learning of equation 3, we compared NHP-U and negative samples chosen uniformly at random. The results for all the undirected hypergraph datasets are shown in tables 8 and 9. In the tables, we have called negative samples chosen uniformly at random

from $\mathcal{E} - E$ as *random negative sampling*. We have called negative samples chosen through positive unlabeled learning of equation 3, i.e. NHP-U, as *positive-unlabeled learning*. Note that the numbers corresponding to this row are the same as those in tables 3 and 4.

In addition to the above two, we used positive unlabeled learning technique of equation 3 to sort the hyperedges (primal) in nondecreasing order of their similarities and then selected uniformly at random from only the first half of the sorted order (i.e. most dissimilar hyperedges). We have called this technique *mixed* as, intuitively, it provides benefits of both positive unlabeled learning and uniform random negative sampling. More principled approaches than *mixed* is left for future work.

### DISCUSSION OF RESULTS

As we can see in table 9, the standard deviations of *random negative sampling* are on the higher side. This is expected as the particular choice made for negative samples decides the decision boundary for the binary classifier. The superior AUC values of *mixed* in table 8 supports our intuition that it provides benefits of both positive unlabeled learning and uniform random negative sampling. The standard deviations of *mixed* are much lower but still higher than *positive-unlabeled learning*. In general, summarising the results for all datasets, we believe that positive-unlabeled learning is superior to *random negative sampling* because of the higher confidence (low standard deviation) predictions.

## 8 CONCLUSION AND FUTURE WORK

We have introduced NHP, a novel neural approach for hyperlink prediction in both undirected and directed hypergraphs. To the best of our knowledge, this is the first neural method for hyperlink prediction in undirected hypergraphs. NHP is also the first method for hyperlink prediction in directed hypergraphs. Through extensive experiments on multiple real-world datasets, we have demonstrated NHP's effectiveness over state-of-the art baselines. Approaches that augment GCNs with attention (Veličković et al., 2018), self-training and co-training with random walks (Li et al., 2018), edge-feature learning in a dual-primal setup (Monti et al., 2018) have been recently proposed on graph-based semi-supervised learning tasks. Our NHP framework provides the flexibility to incorporate these approaches for more improved performance. An interesting future direction is predicting hyperlinks in partial-order hypergraphs (Feng et al., 2018). We leave extending NHP framework to inductive settings as part of future work.

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

# APPENDIX

## 8.1 HYPERPARAMETER DETAILS

We used the same hyperparameters as Kipf & Welling (2017) for the 2-layer GCN model for all the datasets in all the experiments.

Additionally for the 2-layer multi-layer perceptron used for the directed hyperlink prediction experiments, we used 16 hidden units with a dropout rate of 0.25

## COAUTHORSHIP DATASETS CONSTRUCTION DETAILS

- Cora: We used the author data[1] to get the co-authorship hypergraph for cora[2].

---

[1] https://people.cs.umass.edu/ mccallum/data.html
[2] https://linqs.soe.ucsc.edu/data

| hyperparameter | value |
|---|---|
| number of hidden units | 16 |
| number of hidden layers | 2 |
| dropout rate | 0.5 |
| L2 regularisation | $5 \times 10^{-4}$ |
| learning rate | 0.01 |
| non-linearity | ReLU |

Table 10: Hyperparameters of GCN used for all the datasets

- DBLP: We used the DBLP database v4[3]. We filtered out papers without abstracts, and processed each abstract by tokenizing it and removing stop-words removal. Further, we filtered out papers with one author only. This left 540532 papers.

  In order to ensure that the hypergraph formed would be sufficiently dense, we found the number of papers authored by each author and took the top 1000 authors as 'selected authors'. Then we filtered out the papers that were not authored by at least three of the selected authors. Finally, we were left with 1590 papers by 685 of the original 1000 selected authors.

  To extract word features from each of these abstracts, we took all words appearing in these abstracts with a frequency greater than 50. Each abstract was thus represented by a 602-dimensional bag-of-words representation.

For both datasets, we randomly sample $|E|$ fake papers according to the author distribution of the existing non-fake papers (2708 and 1590 for CORA and DBLP respectively). We randomly generated Gaussian $p$ dimensional features for these fake papers (1433 and 602 for CORA and DBLP respectively).

---

[3]https://aminer.org/lab-datasets/citation/DBLP-citation-Jan8.tar.bz2

