# OpenReview forum: "Link Prediction in Hypergraphs using Graph Convolutional Networks"
_ICLR.cc/2019/Conference_

### Official Review · AnonReviewer3 · 2018-10-26
**Novelty of the proposed method is very marginal**

**Rating:** 4
**Confidence:** 5

**Review:**

This paper proposed to use graph convolutional neural networks for link prediction. The authors proposed to use the dual graph to simultaneously learn node and edge embeddings. The label of the edges (positive or negative) are used as supervised signal for training the GCNs. Experiments on a few small data set prove the effectiveness of the proposed approaches.

Strength:
- important problem

Weakness:
- the novelty of the proposed method is very marginal
- the experiments are quite weak

Details:
- the novelty of the proposed method seems to be very marginal, which simply applies the GCN for link prediction. The existing GCN based method for recommendation shares similar ideas (e.g., Yin et al. 2018, PinSage), though dual hypergraph is not used. But the essential idea is very similar.
- the data sets used in the experiments are too small
- the node embedding based methods should be compared for link prediction, e.g., DeepWalk, LINE, and node2vec.

---

> ### Author Response · Authors · 2018-11-10
> **Our response to AnonReviewer3**
>
> Thanks for the review.
>
> On the novelty of our work:
> Link prediction in undirected hypergraphs is an underexplored problem and that in directed hypergraphs is an unexplored problem. Our main contribution is a unified framework for both the settings and our proposed solution is conceptually simple, yet effective. We believe the problem settings are important and interesting (as noted by the other reviewers too), and that this paper will inspire further research in this direction.
>
>
>
> On comparison with PinSage [Ying et al. KDD 2018]:
> PinSage has been designed to work on the bipartite graph of Pinterest. The Pinterest graph can be seen as the star expansion of a hypergraph with pins (hypernodes) on one side of the partition and boards (hyperlinks) on the other side. Following the reviewer’s suggestion, we have compared NHP against star expansion below.
>
>
>
> On comparison with node2vec:
> Following the reviewer’s suggestion, we have compared NHP against node2vec. Node2vec has been shown to be superior to DeepWalk and LINE in [Grover et al. KDD 2016] and hence we have compared only against it. We have also compared NHP against CMM+MLP as suggested by reviewer #2. We report only the number of reactions recovered in the undirected hypergraph experiments.
>
> -----------------------------------------------------------------------------------------------------------------------------------
> dataset				  	       iAF692	           iHN637	               iAF1260b	              iJO1366
> -----------------------------------------------------------------------------------------------------------------------------------
> node2vec 				     299 +/- 10	          303 +/- 4	     1100 +/- 13	          1221 +/- 21
> -----------------------------------------------------------------------------------------------------------------------------------
> GCN on star expansion	            174 +/- 5	         219 +/- 12	      649 +/- 10	           568 +/- 18
> -----------------------------------------------------------------------------------------------------------------------------------
> NHP-U (ours)			            313 +/- 6	          360 +/- 5	     1258 +/- 9	           1381 +/- 9
> -----------------------------------------------------------------------------------------------------------------------------------
>
> We request the reviewer to see the updated paper for AUC numbers. We have updated the results for all the other datasets and experiments and we request the reviewer to see the paper.
>
> From the table above, we can see that the star expansion of a hypergraph is less effective because there are no direct connections between chemical reactions (because the graph is bipartite). Clique expansion, on the other hand, connects two chemical reactions if they share a chemical substance and hence can exploit the relationships much better.
>
>
>
> On the size of the datasets used:
> Our work was motivated by the task of predicting reactions, for which we used datasets already available in the literature (given by Zhang et. al, AAAI 2018). Regarding the co-authorship datasets used, we had to filter the large datasets already available to ensure that meaningful hyperlinks were obtained which led to some reduction in size. We request the reviewer to take a look at the appendix for the exact details.
>
>
>
> On simultaneous learning of node and edge embeddings:
> NHP, our proposed method, learns node embeddings in the dual hypergraph which is the same as learning hyperlink embeddings in the primal.  While PinSage works on the Pinterest bipartite graph (star expansion) and hence involves simultaneous learning of node/edge embeddings, NHP works on the clique expansion and learns node embeddings of the dual.

---

### Official Review · AnonReviewer2 · 2018-10-31
**Interesting and important problem; technical contribution is limited given existing work.**

**Rating:** 5
**Confidence:** 4

**Review:**

This paper proposed Neural Hyperlink Predictor (NHP) to perform link prediction based on graph convolutional network (GCN). Following prior work, the hyperlink prediction is perform in the dual hypergraph, where each node represents a hyperlink in the primal hypergraph. The original problem is then equivalent to a simple node classification problem. To deal with directed hyperlink, a separate term is added to distinguish heads from tails.

The problem of link prediction in hypergraph is important and interesting, especially in the chemistry domain. However from the technical point of view, this work is somewhat incremental since prior work has done link prediction using GCN (Zhang and Chen, 2018). The idea of performing hyperlink prediction in the dual hypergraph is not new, either (Lugo-Martinez and Radivojac, 2017). As for the directed hypergraph setting, it seems to be a straightforward extension once one knows how to do in the undirected setting (adding an extra term to classify head/tail).

In terms of experiments, given the similarity between Lugo-Martinez and Radivojac, 2017 and NHP (both operates in the dual hypergraph), it would be better if the former could also be used as a baseline, as least in the undirected setting.

It is reasonable to have a subset of links as candidate reactions in the metoboli network datasets. For CORA and DBLP, it is not clear where the ‘actual papers’ and ‘candidate papers’ come from. For example in CORA there are 1072 authors; yet there are only 5416 candidate papers.

It seems the joint learning of NHP-D does not improve the accuracy in the directed setting as claimed in Sec. 5.2. Besides, there is no baseline in the directed setting. It is difficult to appreciate the performance in Sec. 6. One thing one can do is to use previous methods in the undirected setting, e.g., CMM, with the extra term L_d in Eq. (4).

Minor comments:
Typo:
P5: atleast -> at least
P5: What is GCN 2?
Sec. 5: ‘p = 32 in 1’ and ‘shown in 2’

Missing references on link prediction and/or deep learning:
Discriminative relational topic models. PAMI 2014.
Relational deep learning: A deep latent variable model for link prediction. AAAI 2017
Neural relational topic models for scientific article analysis. CIKM 2018.

---

> ### Author Response · Authors · 2018-11-10
> **Our response to AnonReviewer2**
>
> Thanks for the review.
>
> On the novelty of our work:
> Link prediction in undirected hypergraphs is an underexplored problem and that in directed hypergraphs is an unexplored problem. Our main contribution is a unified framework for both the settings and our proposed solution is conceptually simple, yet effective. We believe the problem settings are important and interesting (as noted by the other reviewers too), and that this paper will inspire further research in this direction.
>
>
>
> On adding an extra term to CMM as a baseline for directed hyperlink experiments:
> Following the reviewer’s suggestion, we have added CMM + MLP as a baseline. We have also compared NHP against node2vec + MLP and star expansion + MLP as suggested by reviewer #3. We report below the number of reactions recovered in the directed hypergraph experiments.
>
> ---------------------------------------------------------------------------------------------------------------------------------
>                             dataset				iAF692	         iHN637	          iAF1260b          iJO1366
> ---------------------------------------------------------------------------------------------------------------------------------
> node2vec + MLP			                      255 +/- 5	         237 +/- 5	  838 +/- 13	   902 +/- 11
> ---------------------------------------------------------------------------------------------------------------------------------
> CMM + MLP    			                     253 +/- 9	        241 +/- 11	  757 +/- 26	   848 +/- 21
> ---------------------------------------------------------------------------------------------------------------------------------
> GCN on star expansion + MLP	             242 +/- 5	        241 +/- 10	   786 +/- 13	   852 +/- 11
> ---------------------------------------------------------------------------------------------------------------------------------
>
> NHP-D (sequential)			             263 +/- 7	       221 +/- 10	  867 +/- 31	   954 +/- 29
>
> NHP-D (joint)				            262 +/- 8	        236 +/- 8	  869 +/- 13	   944 +/- 20
>
> ---------------------------------------------------------------------------------------------------------------------------------
> We request the reviewer to see the updated paper for AUC numbers. We have also observed that NHP-U outperforms all its baselines in the undirected experiments and the results have been updated in our paper.
>
>
>
> On candidate papers in coauthorship networks such as cora:
> The standard cora dataset has 2708 papers. We sampled an equal number of fake papers at random to get the 5416 candidate papers for cora. We request the reviewer to see the appendix for more details.
>
>
>
> On comparison with Lugo-Martinez and Radivojac, 2017:
> We had difficulty reproducing the results, given that the code was not available and the authors did not respond to our request emails, and the method as described in the paper is rather vague about the details.

---

> > ### Comment · AnonReviewer2 · 2018-12-05
> > **Addressed some of my concerns, baselines need clarification, marginal technical merit**
> >
> > Thanks for your response. It addressed some of my concerns. However, I still have concerns on the baselines in the directed setting and the technical merit (novelty). For the baseline in the directed setting, does the CMM+MLP include the extra term L_d in Eq. (4) or it is just the plain CMM plus MLP?
> >
> > For the sampling of candidate papers, it seems to be quite a unrealistic (and overly simplified) setting, since the number of possible combinations of authors is huge. It would be interesting (and more realistic) to sample much more fake papers and see how the methods perform (ideally we can see similar margin between NHP and the baselines).

---

> > > ### Author Response · Authors · 2018-12-05
> > > **Our clarifications**
> > >
> > > Thanks for the response.
> > >
> > > On the novelty of our work:
> > > We reiterate that the main novelty / contribution of our work is to explore
> > > 1) an unexplored problem (link prediction in directed hypergraphs)
> > > 2) an underexplored problem (link prediction in undirected hypergraphs) and to propose the first neural-network-based method for the problem
> > >
> > > We have proposed a unified framework for the two important and interesting problems and our proposed solution is conceptually simple, yet effective.
> > >
> > >
> > >
> > > On including the extra term L_d for CMM + MLP:
> > > The baseline we have compared against is plain CMM + MLP (sequential). CMM uses the expectation-maximisation (EM) algorithm to optimise its objective function to predict hyperlinks. Since CMM is not solved by the conventional gradient descent-based methods, using the term L_d jointly with EM is a non-trivial problem in itself (it is not as straightforward as adding an extra term to the loss function).
> > >
> > >
> > >
> > > On sampling candidate papers:
> > > As motivated in section 3, in the case of multi-author collaborations of academic/technical papers, hyperlinks have cardinalities less than a small number, as papers seldom have more than 6 authors. We looked at the distribution of the number of authors of actual (positive) papers and sampled an equal number of negative (fake) papers from the distribution. This means that although there are a large number of potential fake papers, we can make do with a vastly reduced number because of our sampling strategy.
> > >
> > > A related work [1] also has sampled an equal number of negative links for all datasets in its experiments.
> > > [1] Link Prediction Based on Graph Neural Networks, Muhan Zhang and Yixin Chen, NeurIPS 2018

---

### Official Review · AnonReviewer1 · 2018-11-02
**Interesting problem, but incremental contribution**

**Rating:** 6
**Confidence:** 2

**Review:**

[Relevance] Is this paper relevant to the ICLR audience? yes

[Significance] Are the results significant? somewhat

[Novelty] Are the problems or approaches novel? rather incremental

[Soundness] Is the paper technically sound? yes

[Evaluation] Are claims well-supported by theoretical analysis or experimental results? marginal

[Clarity] Is the paper well-organized and clearly written? okay

Confidence: 2/5

Seen submission posted elsewhere: No

Detailed comments:

In this work, the authors propose an approach to the (hyper-) link prediction problem in both directed and undirected hypergraphs. The approach first applies an existing dual transformation to the hypergraph such that the link prediction problem (in the primal) becomes a node classification problem in the dual. They then use GCNs to classify the (dual) nodes. Experimentally, the proposed approach marginally outperforms existing approaches.

=== Major comments

I found the novelty of the proposed approach rather limited. The proposed approach essentially just concatenates three existing strategies (dual reformulation from Scheinerman and Ullman, GCNs from Kipf and Welling, and negative sampling which is common in many communities, e.g., Han and Chen, but many others, as well). I believe the contribution for link prediction in directed hypergraphs is a more novel contribution, however, I had difficulty following that discussion.

It is difficult to interpret the experimental results. Tables 3 and 6 do not include a measure of variance. Thus, it is not clear if any of the results are statistically significant. It is also not clear whether the “10 trials” mentioned in the figure captions correspond to a 10-fold cross-validation scheme or something else. It is unclear to me what the random feature matrix for the metabolic network is supposed to me or do. It is also unclear to me why “fake papers” are needed for the citation networks; it is clear that “fake author lists” are needed for negative sampling, but it seems they could be attached to existing papers. Similarly, it is unclear how the set of candidate edges (\mathcal{E}) was chosen.

I appreciate that the authors made the code available. I did not run it, but I did have a look, and I believe it could be adapted by others without an unreasonable amount of work.

=== Minor comments

This work is very similar to the arXiv submission 1809.09401. To the best of my knowledge, though, that work has not yet been published in a peer-reviewed venue, so I do not consider it a problem that it is not cited here.

According to Tables 1 and 2, iAF692 and iHN637 datasets are smaller than the other datasets except DBLP; those two are also less dense than DBLP. According to Table 3, NHP-U seems noticeably better than SHC and CMM on the, while does not appear very significant in the other cases. Is there some relationship between NHP’s performance and the size/density of the graph? or is there some other explanation for this behavior?

Related to the above point, Table 3 shows that the performance on the undirected versions for those two datasets is better than on the other two metabolic networks, while Table 6 shows the opposite for the directed versions. Is there some explanation for this? For example, are there qualitative differences in the size of the hypernodes?

The described strategy for negative sampling seems as though it selects “easy” negative samples, in the sense that they are far away from observed positives; thus, they are also likely far away from any sort of decision boundary. How does the performance change if more “difficult” (or just uniformly random) negative samples are chosen?

I believe Recall@100 (or Precision@100, or @$\Delta E$, etc.) is a more meaningful value to report in Tables 4 and 7, rather than the raw number of edges. That is, it would be more helpful to report something so that numbers across datasets are at least somewhat comparable.

=== Typos, etc.

In Equation (4), the “k” index in d_{ijk} is in {1,2}, but in the text, it is in {0,1}.

“table 2” -> “Table 2”, and many other similar examples throughout the paper.

“higher-order etc.” -> “higher-order, etc.”
“GCN based” -> “GCN-based”, and similar in several places in the paper
“a incomplete” -> “an incomplete”

---

> ### Author Response · Authors · 2018-11-10
> **Our response to major comments of AnonReviewer1**
>
> Thanks for the review
>
> On the novelty of our work:
> Link prediction in undirected hypergraphs is an underexplored problem and that in directed hypergraphs is an unexplored problem. Our main contribution is a unified framework for both the settings and our proposed solution is conceptually simple, yet effective. We believe the problem settings are important and interesting (as noted by the other reviewers too), and that this paper will inspire further research in this direction.
>
>
>
> On the discussion of results for directed hyperlink prediction:
> Both NHP-D (joint) and NHP-D (sequential) perform similarly. To appreciate the results, we have added three baselines as suggested by reviewers 2 and 3. We request the reviewer to see below for a sample of the updated results and the paper for all updated results.
>
> ---------------------------------------------------------------------------------------------------------------------------------
>                          dataset				  	iAF692	         iHN637	          iAF1260b          iJO1366
> ---------------------------------------------------------------------------------------------------------------------------------
> node2vec + MLP			                      255 +/- 5	         237 +/- 5	  838 +/- 13	   902 +/- 11
> ---------------------------------------------------------------------------------------------------------------------------------
> CMM + MLP    			                     253 +/- 9	        241 +/- 11	  757 +/- 26	   848 +/- 21
> ---------------------------------------------------------------------------------------------------------------------------------
> GCN on star expansion + MLP	             242 +/- 5	        241 +/- 10	   786 +/- 13	   852 +/- 11
> ---------------------------------------------------------------------------------------------------------------------------------
>
> NHP-D (sequential)			             263 +/- 7	       221 +/- 10	  867 +/- 31	   954 +/- 29
>
> NHP-D (joint)				            262 +/- 8	        236 +/- 8	  869 +/- 13	   944 +/- 20
>
> ---------------------------------------------------------------------------------------------------------------------------------
>
>
>
> On variance in the results:
> We observed variances of AUC values to be in the third decimal places (i.e., very close to zero). We have reported variances in the number of hyperlinks recovered in all experiments. These are much more interpretable/statistically significant.
>
>
>
> On 10 trials:
> We report the mean values over 10 different splits of train and test.
>
>
>
> On random features:
> The feature initialisations are random for metabolic network experiments as we do not have any available features to exploit. We believe the neighbourhood feature aggregation of GCN causes useful node embeddings to be learnt during training.
> We also observe that NHP is competitive with a node2vec baseline (suggested by reviewer 3) which is a featureless approach.
>
>
>
> On creation of fake papers:
> In these experiments, authors correspond to nodes in the (primal) graph, while papers correspond to hyperlinks, i.e., sets of authors. So in this context, fake papers are the same as fake author lists and hence cannot be attached to existing (true) papers. The set of candidate edges is the set of true papers union the set of fake papers.

---

> > ### Author Response · Authors · 2018-11-22
> > **Our response to minor comments of AnonReviewer1**
> >
> > On comparisons with random negative sampling:
> > Below, we have compared our strategy of positive-unlabeled learning against uniform random negative sampling.
> > -------------------------------------------------------------------------------------------------------------------------
> >                   dataset                           iAF692        iHN637           iAF1260b          iJO1366
> > -------------------------------------------------------------------------------------------------------------------------
> > random negative sampling        236 +/- 32     415 +/- 47     967 +/- 125     1074 +/- 168
> > -------------------------------------------------------------------------------------------------------------------------
> > positive-unlabeled learning         313 +/- 6     360 +/- 5        1258 +/- 9         1381 +/- 9
> > -------------------------------------------------------------------------------------------------------------------------
> >
> > As can be seen from the table, the standard deviations of random negative sampling are on the higher side. This is expected as the particular choice made for negative samples decides the decision boundary for the binary classifier.
> >
> > We request the reviewer to see the updated paper for AUC numbers and a discussion around these results in the updated section 7 of our paper.
> >
> >
> >
> > On adding Recall@$\Delta E$ in tables:
> > We have added recall@$\Delta E$ of NHP for all datasets in both undirected and directed hypergraph experiments in our updated paper. We have retained the raw hyperlinks recovered as they contain standard deviations in addition to mean values.
> >
> >
> >
> > On the arXiv submission 1809.09401:
> > The submission uses the clique expansion to approximate the hypergraph which is similar to our work. We have cited the submission in our updated paper. However, it does not use the dual hypergraph idea nor any negative sampling technique.
> >
> >
> >
> > On connecting experimental results and dataset sizes/densities:
> > We cannot draw general conclusions connecting results and dataset sizes/densities. In general we observe that NHP outperforms the baselines because the graph convolutional network is tailor-made for semi-supervised learning with small amounts of labeled data (10% in our experiments).

---

> > > ### Comment · AnonReviewer1 · 2018-11-24
> > > **Improved experimental results, but novelty still somewhat questionable**
> > >
> > > Hi,
> > >
> > > I have read the authors' responses to my comments, as well as the other reviews and responses. The additional experiments do clarify some of my questions about the empirical behavior of the proposed approach.
> > >
> > > Still, all reviewers had questions about the novelty of the proposed approach. I do not believe any of the rebuttals address these concerns, so my overall rating is not changed.

---

> > > > ### Author Response · Authors · 2018-12-05
> > > > **On the novelty of our work**
> > > >
> > > > Thanks for the response. We reiterate that the main novelty / contribution of our work is to explore
> > > > 1) an unexplored problem (link prediction in directed hypergraphs)
> > > > 2) an underexplored problem (link prediction in undirected hypergraphs) and to propose the first neural-network-based method for the problem
> > > >
> > > > We have proposed a unified framework for the two important and interesting problems and our proposed solution is conceptually simple, yet effective.

---

### Author Response · Authors · 2018-11-22
**Summary of revisions**

We thank the reviewers for their reviews. Below, we have summarised the revisions made to our paper in the rebuttal period. The majority of the revisions have been in the experiments (section numbers 5, 6 and 7).

- In section 5, we have added descriptions and results of a couple of baselines (node2vec, and GCN on star expansion) as suggested by reviewer 3.

- In section 6, we have added descriptions and results of three baselines (node2vec + MLP, CMM + MLP, and GCN on star expansion + MLP) as suggested by reviewers 2  and 3.

- We have added a new section (section 7) to compare our strategy of positive unlabeled learning and negative sampling uniformly at random as suggested by reviewer 1.



We have corrected all typos, and cited missing references as suggested by the reviewers. The revisions can be compared using the compare revisions option on the revisions page.

---

### Meta-Review · Area_Chair1 · 2018-12-17
**Interesting and important problem, somewhat limited novelty of the approach**

**Confidence:** 4
**Recommendation:** Reject

**Metareview:**

The paper describes  a method for the link prediction problem in both directed and undirected hypergraphs.  While the problem discussed in the paper is clearly importnant and interesting, all reviewers agree that the novelty of the proposed approach is somewhat limited given the prior art.